# Phase-Space Methods for Simulating the Dissipative Many-Body Dynamics of Collective Spin Systems

J. Huber[1*], P. Kirton[1,2], P. Rabl[1],

**1** Vienna Center for Quantum Science and Technology, Atominstitut, TU Wien, 1040 Vienna, Austria
**2** Department of Physics and SUPA, University of Strathclyde, Glasgow G4 0NG, UK
\* julian.huber@tuwien.ac.at

May 18, 2021

## Abstract

We describe an efficient numerical method for simulating the dynamics and steady states of collective spin systems in the presence of dephasing and decay. The method is based on the Schwinger boson representation of spin operators and uses an extension of the truncated Wigner approximation to map the exact open system dynamics onto stochastic differential equations for the corresponding phase space distribution. This approach is most effective in the limit of very large spin quantum numbers, where exact numerical simulations and other approximation methods are no longer applicable. We benchmark this numerical technique for known superradiant decay and spin-squeezing processes and illustrate its application for the simulation of non-equilibrium phase transitions in dissipative spin lattice models.

# 1   Introduction

Large ensembles of two-level systems that can be approximately modeled as a single collective spin are of interest in many areas of physics. In quantum optics, collective light-matter interaction effects can be understood from the analysis of the Dicke model [1], which describes the coupling of many two-level atoms to a common photonic mode. In the field of ultracold atoms, the evolution of Bose-Einstein condensates in double-well potentials can be mapped onto the motion of one large collective spin [2]. In nuclear and solid-state physics the Lipkin-Meshkov-Glick (LMG) model [3] is frequently used to investigate ferromagnetic phase transitions in systems with all-to-all interactions. In many situations one is interested in the dynamics or the steady states of those systems in the presence of dephasing and decay. For example, for magnetic field sensing, spin squeezing [4,5] and related metrological applications with large ensembles of atoms [6,7] the achievable sensitivities are primarily limited by such decoherence processes. But the coupling of large ensembles of two-level systems to a common environment can also lead to new physical phenomena, such as phase-locked condensates in equilibrium [8], or superradiant [1,9] and super-correlated [10] decay.

From a theoretical and computational perspective, the primary interest in collective spin models arises from their permutational symmetry. This symmetry effectively reduces the full dimension of the Hilbert space of $\mathcal{N}_{\text{TLS}}$ two-level systems, $d = 2^{\mathcal{N}_{\text{TLS}}}$, to the dimension $d_S = (2S + 1)$ of a spin $S = \mathcal{N}_{\text{TLS}}/2$ system. At the same time, the system can still exhibit interesting many-body effects and sharp phase transitions in the 'thermodynamic limit' $S \gg 1$. For this reason, dissipative versions of the Dicke [11–13], LMG [14,15] and related collective spin models [16–18] play an important role in the analysis of non-equilibrium phase transitions in open quantum systems, since an exact numerical integration of the full master equation is still possible for moderately large ensembles. However, brute force numerical simulations are no longer feasible for atom numbers encountered in many of the actual experiments and, in general, for systems involving multiple collective spins. Such scenarios appear naturally in the presence of inhomogeneous couplings or frequencies [19, 20] or in extended lattice systems [19, 21], which are most relevant for analyzing critical phenomena. For closed systems, the coherent dynamics of spins with $S \gg 1$ is typically well-described by mean-field theory, i.e., by evaluating the dynamics of the average spin vector $\langle \vec{S} \rangle$ only. But this approach ignores important correlations, such as spin squeezing effects, and will in general provide a poor description of the actual state in the presence of dissipation. Here quantum fluctuations associated with incoherent processes can drive the system into highly mixed states [16, 17, 21], where the fluctuations of the spin components become comparable to their mean values. Thus, in order to accurately model such a behavior, it requires *approximate*

numerical techniques, which take the effect of fluctuations into account, while still being able to simulate the dynamics of large collective spins efficiently.

In this paper we describe a broadly applicable stochastic method for simulating the dissipative dynamics of systems involving either a single or many coupled collective spins. The method relies on the Schwinger boson representation of spin systems and uses an extension of the truncated Wigner approximation (TWA) [22] to map the dynamics of those bosons onto a set of stochastic differential equations in phase space. For weakly interacting bosonic systems such phase space methods based on the TWA are well established and can be used, for example, to simulate dissipative bosonic lattice systems and non-equilibrium condensation phenomena [23–27]. The extension of these methods to spin systems via the Schwinger representation has previously been applied for simulating the coherent dynamics of lattices of spin-1/2 systems [28, 29] and that of collective spins [22]. In the latter case also alternative methods based on the discrete TWA (DTWA) [30] are very efficient. In Ref. [31], Olsen *et al.* showed that a stochastic sampling of the positive-$P$ representation of the Schwinger bosons can also be used to model the collective decay of an atomic ensemble. However, the practical applications of this approach are very limited since the stochastic trajectories derived from the positive-$P$ distribution tend to diverge after rather short times [28, 29, 31, 32] and, to our knowledge, this method has not been developed further. Here we show how this problem can be overcome for systems with $S \gg 1$ by working with the Wigner function, but performing an additional positive diffusion approximation (PDA). As a result of this approximation, the stochastic equations in phase space are well-behaved for arbitrary times, which allows us to evaluate also the long-time dynamics and the steady states of dissipative spin systems that have been inaccessible so far. Using this truncated Wigner method for open quantum spins (TWOQS), we obtain an approximately linear scaling with the number of collective spins included, in any dimension and for arbitrary interaction patterns. In a recent work [21], we have already applied this method to identify novel $\mathcal{PT}$-symmetry breaking transitions in the steady state of a one dimensional spin lattice with gain and loss. Here we provide a more detailed derivation of this simulation technique and discuss and benchmark its performance in terms of several explicit examples.

The structure of this paper is as follows: In Sec. 2 we first present a general outline of the method and explain how the original master equation can be mapped, under certain approximations, onto a set of stochastic differential equations. Then, in Sec. 3 we illustrate and benchmark this procedure by studying several model systems. We also go on and show how this technique can be applied to the simulation of non-equilibrium phase transitions in dissipative spin lattice models. Finally, in Sec. 4 we present our conclusions.

## 2   Outline of the Method

We are interested in the open system dynamics of $i = 1, \ldots, N$ coupled spin-$S$ systems, which can be modeled by a master equation for the system density operator $\rho$,

$$\dot{\rho} = -i[H, \rho] + \sum_n \Gamma_n \mathcal{D}[c_n]\rho. \tag{1}$$

Here $H$ is the many-body Hamiltonian describing the coherent evolution and the Lindblad superoperators, where $\mathcal{D}[c]\rho = 2c\rho c^\dagger - c^\dagger c\rho - \rho c^\dagger c$, account for incoherent processes with jump operators $c_n$ and rates $\Gamma_n$. In the following we assume that $H$ and all $c_n$ can be written in

terms of products of the collective spin operators $S_i^z$ and $S_i^{\pm} = (S_i^x \pm i S_i^y)$, which obey the usual spin commutation relations, $[S_i^z, S_j^+] = \delta_{ij} S_i^+$ and $[S_i^+, S_j^-] = 2\delta_{ij} S_i^z$.

Equation (1) conserves the length of each individual spin, $\partial_t \langle \vec{S}_i^2 \rangle = 0$, and therefore the dynamics of each subsystem can be restricted to a $d_S = (2S + 1)$ dimensional subspace. However, the dimension of the full density operator, $d_\rho = (d_S)^{2N}$, still scales exponentially with the number of subsystems or lattices sites $N$. This scaling makes an exact numerical integration of Eq. (1) impossible when $S$ or $N$ are large. Here we introduce an approximate method, the TWOQS, to simulate such systems in the limit $S \gg 1$, which only scales linearly with the system size $N$. The derivation of this method consists of four main steps:

1. The $N$ spins are mapped onto a set of $2N$ bosonic modes using the Schwinger boson representation.

2. The master equation for the bosons is mapped onto an equivalent partial differential equation for the Wigner phase-space distribution.

3. We use the TWA and the PDA to obtain a Fokker-Planck equation (FPE) for the Wigner function with an (almost) positive diffusion matrix.

4. This FPE is mapped onto an equivalent set of stochastic Ito equations, which can be efficiently simulated numerically.

In the following, we first give a brief general outline of the individual steps in this derivation, while the application of this method for concrete examples is discussed in more detail in Sec. 3.

## 2.1 Bosonization

In a first step, we use the Schwinger boson representation to map each of the spins, $\vec{S}_i$, onto two independent bosonic modes, $a_i$ and $b_i$, by identifying

$$S_i^+ = a_i^\dagger b_i, \qquad S_i^- = a_i b_i^\dagger, \qquad S_i^z = \frac{1}{2}(a_i^\dagger a_i - b_i^\dagger b_i). \qquad (2)$$

One can easily show that this transformation preserves all the spin commutation relations given above. For all models constructed from collective spin operators only, the total number of excitations at each site, $a_i^\dagger a_i + b_i^\dagger b_i$, is conserved. The initial condition can then be chosen such that

$$\frac{1}{2}(a_i^\dagger a_i + b_i^\dagger b_i) = S \qquad (3)$$

to simulate spins of different lengths. This is more useful than mapping each site to a single Holstein-Primakoff boson [33], since the transformation above does not involve any operator square roots, which can be numerically difficult to work with.

## 2.2 Phase Space Distributions

The main advantage of switching to a representation expressed in terms of bosonic modes is that the master equation, Eq. (1), can be mapped onto an equivalent partial differential equation for a class of phase space distributions, which contain the same information as the

density operator [32]. We parameterize the set of distributions by the variable $k = -1, 0, 1$ and define

$$F_k(\vec{\alpha}, t) = \frac{1}{\pi^{4N}} \int d^{4N}\lambda \, e^{(\vec{\alpha}\vec{\lambda}^* - \vec{\alpha}^*\vec{\lambda})} \, \mathrm{Tr}\left\{ e^{\vec{\lambda}\vec{v}^\dagger} \rho(t) e^{-\vec{\lambda}^*\vec{v}} \right\} e^{\frac{(1+k)}{2}|\vec{\lambda}|^2}, \tag{4}$$

where $\vec{v} = (a_1, b_1, a_2, b_2, \ldots, a_N, b_N)$ is a vector of all $2N$ bosonic annihilation operators and $\vec{\alpha}$ and $\vec{\lambda}$ are vectors containing the same amount of complex numbers. When $k = 0$ this phase space distribution corresponds to the Wigner function, for $k = 1$ it is the Glauber-Sudarshan $P$-representation and when $k = -1$ we obtain the Husimi $Q$-function. We can use this definition to calculate what form each term in the master equation takes in the equation for $F_k$ [32]. For example, for a single mode one finds the mapping

$$a\rho \;\rightarrow\; \left[ \alpha + \frac{(1-k)}{2} \frac{\partial}{\partial\alpha^*} \right] F_k(\alpha, t), \tag{5}$$

$$a^\dagger\rho \;\rightarrow\; \left[ \alpha^* - \frac{(1+k)}{2} \frac{\partial}{\partial\alpha} \right] F_k(\alpha, t), \tag{6}$$

$$\rho a^\dagger \;\rightarrow\; \left[ \alpha^* + \frac{(1-k)}{2} \frac{\partial}{\partial\alpha} \right] F_k(\alpha, t), \tag{7}$$

$$\rho a \;\rightarrow\; \left[ \alpha - \frac{(1+k)}{2} \frac{\partial}{\partial\alpha^*} \right] F_k(\alpha, t). \tag{8}$$

This translation lets us recast the master equation for $\rho$ in the form of a partial differential equation for the phase space distribution,

$$\frac{\partial}{\partial t} F_k(\vec{\alpha}, t) = L F_k(\vec{\alpha}, t), \tag{9}$$

with $L$ some linear differential operator that depends on the specific problem under consideration.

## 2.3 Truncated Wigner Approximation

The result in Eq. (9) is still exact and therefore in general not very useful. In particular, the differential operator $L$ may contain third- or higher-order derivatives, which prevent an efficient stochastic sampling of $F_k$. For example, for the coherent dynamics generated by the master equation $\dot{\rho} = -i\Omega[S_x^2, \rho]$, the corresponding partial differential equation for $F_k \equiv F_k(\alpha, \beta, t)$ reads

$$\frac{\partial F_k}{\partial t} = \frac{i\Omega}{4} \left[ \frac{\partial}{\partial\alpha} \left( 2\alpha^*\beta^2 + 2\alpha|\beta|^2 \right) + \frac{\partial}{\partial\beta} \left( 2\beta^*\alpha^2 + 2|\alpha|^2\beta \right) - k\frac{\partial^2}{\partial\alpha^2}\beta^2 - k\frac{\partial^2}{\partial\beta^2}\alpha^2 - k\frac{\partial^2}{\partial\alpha\beta}\alpha\beta \right.$$
$$\left. + \frac{1-k^2}{2} \left( \frac{\partial^3}{\partial\alpha\partial\alpha^*\partial\beta}\beta - \frac{\partial^3}{\partial\beta\partial\beta^*\partial\alpha}\alpha - \frac{\partial^3}{\partial\alpha^2\partial\beta^*}\beta - \frac{\partial^3}{\partial\alpha^*\partial\beta^2}\alpha \right) - c.c. \right] F_k. \tag{10}$$

To proceed we neglect all third- and higher-order derivatives, which in this example corresponds to omitting all terms in the second line of Eq. (10). This approximation is just the usual TWA [22] applied to arbitrary distribution functions. For spin systems we expect this approximation to become accurate in the limit of large $S$, since terms proportional to $\alpha F_k$ or

$\beta F_k$ scale as $\sim \sqrt{S}$ compared to derivatives such as $\partial F_k / \partial \alpha \sim O(1)$. After performing the TWA we obtain a FPE of the form

$$\frac{\partial}{\partial t} F_k(\vec{x}, t) = \left[ -\frac{\partial}{\partial x_j} A_j(\vec{x}) + \frac{1}{2} \frac{\partial}{\partial x_i} \frac{\partial}{\partial x_j^*} D_{ij}(\vec{x}) \right] F_k(\vec{x}, t), \tag{11}$$

with a drift matrix $A$ and a diffusion matrix $D$. Here we have assumed Einstein's sum convention, where the indices $i$ and $j$ run over the $4N$ components of the vector $\vec{x} = (\alpha_1, \alpha_1^*, \beta_1, \beta_1^*, \alpha_2, \alpha_2^*, \beta_2, \beta_2^*, \dots)$.

## 2.4 Positive Diffusion Approximation

For stochastic simulations, performing the TWA is not enough since in general the diffusion matrix $D$ obtained in this way is not positive semi-definite. This can already be seen from the underlined terms in Eq. (10). Similarly, we find that an incoherent decay process, $\dot{\rho} = \Gamma \mathcal{D}[S^-]\rho$, is mapped under the TWA onto the FPE

$$\frac{\partial}{\partial t} F_k = \Gamma \left[ \frac{\partial}{\partial \alpha} \left( |\beta|^2 + \frac{(1+k)}{2} \right) \alpha - \frac{\partial}{\partial \beta} \left( |\alpha|^2 - \frac{(1-k)}{2} \right) \beta - \frac{\partial^2}{\partial \alpha \partial \beta} \alpha \beta \right.$$
$$\left. + \frac{(1-k)}{2} \frac{\partial^2}{\partial \alpha \partial \alpha^*} \left( |\beta|^2 + \frac{(1+k)}{2} \right) + \frac{(1+k)}{2} \frac{\partial^2}{\partial \beta \partial \beta^*} \left( |\alpha|^2 - \frac{(1-k)}{2} \right) + c.c. \right] F_k, \tag{12}$$

and there are again second-order derivatives that can lead to negative diffusion rates. Thus, in a second step we perform a PDA by neglecting some of these diffusion terms. In the two examples above this approximation amounts to omitting all the underlined terms in Eq. (10) and Eq. (12), while keeping the diffusion terms in the second line of Eq. (12). This choice cannot be justified by simple scaling arguments and in Sec. 3 we discuss and verify the applicability of this approximation in terms of several explicit examples. In general, the PDA can be motivated by the fact that it eliminates the dominating negative contributions to $D$, while conserving the total spin $S$ and leaving the equations of motion for the mean values $\langle S^k \rangle$ unaffected. The price we pay for this last requirement is that for $k = 0$ the resulting diffusion matrix can become negative for certain values of $\alpha$. However, the corrections scale as $\sim 1/S$ compared to other terms and for $S \gg 1$ the residual negative contributions do not affect considerably the stochastic sampling of trajectories in actual simulations.

Before we proceed let us remark that the problem of non-positivity can also be overcome by working with a positive-$P$ representation, where $\alpha_i$ and $\alpha_i^*$ are replaced by a pair of independent complex variables [31, 32]. In this case, a positive semi-definite diffusion matrix can be obtained for this larger set of variables without neglecting any terms. However, it is known that the resulting stochastic equations are often not well-behaved [32]. In particular, the appearance of "spikes", where individual trajectories diverge at a finite time [28, 29, 32], often prevents the simulation of the long-time behavior of a system or its steady state.

## 2.5 Stochastic Simulations

After applying the TWA and the PDA, we end up with a FPE with an (almost) positive semi-definite diffusion matrix $D$. This FPE can be mapped onto an equivalent set of stochastic (Ito) differential equations [34],

$$dx_i = A_i(\vec{x})dt + B_{ij}(\vec{x})dW_j(t), \tag{13}$$

where $dW_i$ are real-valued independent Wiener processes with $\langle dW_i dW_j \rangle = \delta_{ij} dt$ and $B(\vec{x})$ is the factorized diffusion matrix with $B(\vec{x})B(\vec{x})^\dagger = D(\vec{x})$. This set of equations can be efficiently simulated with the Euler-Maruyama method [34]. This means that we do not calculate the full probability distribution, but instead obtain the required expectation values by averaging over $n_{\text{traj}}$ trajectories of these stochastic equations. Note that for a closed system, where all second- and higher-order derivatives have been neglected, the amplitudes $\alpha_i$ evolve according to the mean-field equations of motion,

$$\dot{x}_i = A_i(\vec{x}), \tag{14}$$

consistent with the usual applications of the TWA [22]. In the presence of dephasing or decay our approach accounts for the corresponding damping terms and the associated amount of quantum fluctuations in a consistent manner.

For sufficiently many trajectories and with initial values sampled according to the distribution $F_k(\vec{\alpha}, t = 0)$, these stochastic averages provide accurate approximations of the corresponding quantum mechanical expectation values

$$\langle (a_i^\dagger)^n a_j^m \rangle_{P|Q|W} = \int d^{4N}\alpha \, (\alpha_i^*)^n \alpha_j^m F_k(\vec{\alpha}, t) \approx \langle (\alpha_i^*)^n \alpha_j^m \rangle_{\text{stoch}}(t), \tag{15}$$

where, depending on the chosen distribution function, $\langle \dots \rangle_{P|Q|W}$ denotes the normally-ordered, anti-normally-ordered or symmetrically-ordered expectation value. All expectation values of the original spin system can then be obtained using the relations in Eq. (2).

### 2.5.1 Initial conditions

In many situations of interest the initial state can be chosen as a fully polarized state with $\langle S_i^z \rangle = -S$ at each site. This corresponds to a state where one of the two Schwinger bosons is prepared in the vacuum state $|0\rangle$, the other one in the Fock state $|2S\rangle$. For $k = -1$ this state is described by the $Q$-function

$$Q_0(\alpha, \beta) = \frac{1}{\pi^2} e^{-(|\alpha|^2 + |\beta|^2)} \frac{|\beta|^{4S}}{(2S)!}, \tag{16}$$

which is positive everywhere and can be used as an initial probability distribution for the trajectories. For $k = 1$ and $k = 0$ the corresponding $P$- and Wigner distributions for Fock states are singluar or have negative values. It is thus necessary to approximate the initial state by replacing the Fock state $|2S\rangle$ by a coherent state with the same mean amplitude. The corresponding initial conditions are then given by

$$P_0(\alpha, \beta) = \delta(\alpha)\delta(\beta - \sqrt{2S}), \tag{17}$$

and

$$W_0(\alpha, \beta) = \frac{4}{\pi^2} e^{-2(|\alpha|^2 + |\beta - \sqrt{2S}|^2)}, \tag{18}$$

respectively. This approximation introduces an uncertainty in the spin quantum number $S$, which, however, scales only with $\sqrt{S}$ and becomes negligible in the limit of interest, $S \gg 1$.

In order to initialize the system in an arbitrary spin coherent state $|\theta, \phi\rangle$ on the Bloch sphere we can simply rotate this state by the angle $\theta$ around the $y$-axis and $\phi$ around the

$z$-axis. This amounts to replacing $\alpha$ and $\beta$ by the rotated amplitudes

$$
\begin{aligned}
\tilde{\alpha} &= e^{i\phi}(\cos(\theta/2)\alpha - \sin(\theta/2)\beta), &\quad (19)\\
\tilde{\beta} &= \sin(\theta/2)\alpha + \cos(\theta/2)\beta, &\quad (20)
\end{aligned}
$$

i.e., $W_{\theta,\phi}(\alpha, \beta) = W_0(\tilde{\alpha}, \tilde{\beta})$.

## 2.6   $P$-, $Q$-, or Wigner Distribution?

Up to now we have kept our analysis completely general and derived all the results for arbitrary distribution functions $F_k(\vec{\alpha})$. This raises the question of which distribution to choose in an actual simulation? It is well-known that squeezed states, which appear commonly in interacting spin systems, cannot be represented by a positive and non-singular $P$-distribution and thus cannot be simulated via the stochastic equations given in Eq. (13) when k=1. The $Q$-distribution ($k = -1$) has the obvious advantage that it can represent spin states with a well-defined spin quantum number, i.e., there is no need to approximate the initial state. Further, as can be seen from Eq. (12), after the PDA the diffusion matrix for the $Q$-distribution is strictly positive semi-definite. However, it turns out that for models that include $(S^x)^2$ or similar interaction terms in the Hamiltonian, performing the PDA eliminates relevant contributions to the coherent dynamics. As can be seen from Eq. (10), this is not the case for the Wigner distribution ($k = 0$), since there are no second-order derivatives in the Hamiltonian dynamics and the PDA only affects incoherent processes. This is true for all quadratic coupling terms in the Hamiltonian $\sim S_i^{\nu} S_j^{\mu}$ ($\nu, \mu = z, \pm$), which already includes the most common types of spin-spin interactions. Therefore, while below we will also discuss several basic examples where the $P$- or the $Q$-distribution yield equally accurate results, we find that for generic interacting systems it is necessary to work with the Wigner function, which reproduces most accurately the Hamiltonian part of the dynamics.

# 3   Examples and Applications

In this section we will present several explicit examples, to show how our method can be applied to simulate some of the most frequently encountered interactions and decoherence processes. To do so we will mainly focus on systems with a single collective spin, where all the results can still be compared with exact numerical results. This will allow us to test the validity of the approximations described above and make a comparison of how the different phase space representations perform under different circumstances. In Sec. 3.5 we will then extend these results and discuss the simulation of a whole chain of collective spins, for which exact methods are no longer available.

## 3.1   Spontaneous Emission

As a first example we consider the collective decay of a large ensemble of two-level systems, which can be described by the master equation

$$
\dot{\rho} = \frac{\Gamma}{2S}\mathcal{D}[S^-]\rho. \qquad (21)
$$

Note that here we rescale the emission rate by a factor $2S$ in order to obtain the same time scale for the dynamics for different values of $S$. After performing the TWA the resulting FPE

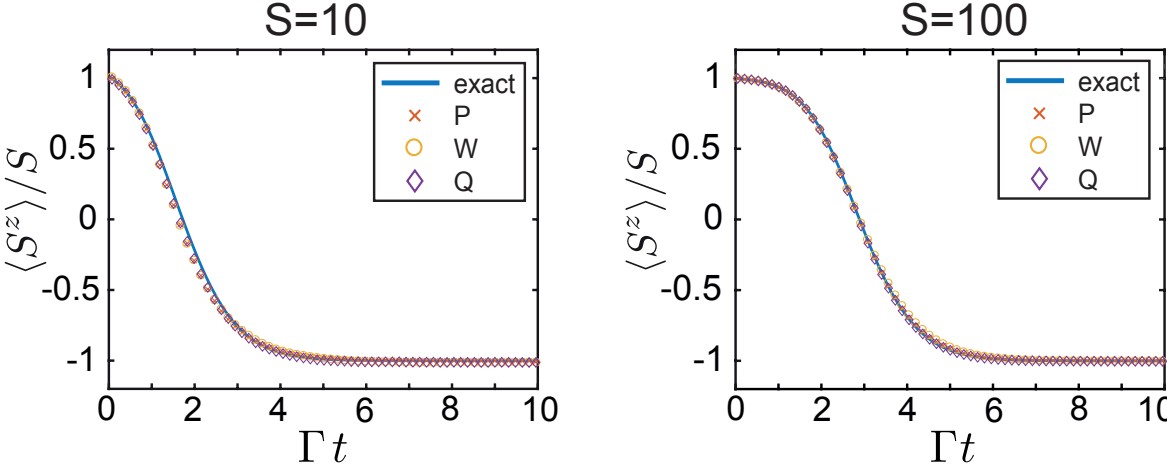

Figure 1: Simulation of the superradiant decay of a single collective spin with spin quantum number $S = 10$ and $S = 100$. The system is initially prepared in the highest excited state, $|S^z = S\rangle$. The stochastic simulations for the $P$-, $Q$- and Wigner-distribution are compared to the exact integration of the master equation, Eq. (21). In both plots $n_{\text{traj}} = 1000$ trajectories have been simulated to compute the stochastic averages.

for this model is already given in Eq. (12) above. The PDA then corresponds to neglecting the underlined term in this equation, after which we can map the FPE onto the following set of stochastic Ito equations

$$d\alpha = \frac{-\Gamma}{2S}\left(|\beta|^2 + \frac{(1+k)}{2}\right)\alpha dt + \sqrt{\frac{\Gamma(1-k)}{4S}\left(|\beta|^2 + \frac{(1+k)}{2}\right)}(dW_1 + idW_2), \quad (22)$$

$$d\beta = \frac{\Gamma}{2S}\left(|\alpha|^2 - \frac{(1-k)}{2}\right)\beta dt + \sqrt{\frac{\Gamma(1+k)}{4S}\left(|\alpha|^2 - \frac{(1-k)}{2}\right)}(dW_3 + idW_4), \quad (23)$$

where the $dW_n$ are real-valued and independent Wiener processes with $\langle dW_n dW_m\rangle = \delta_{nm}dt$.

In Fig. 1 we plot the outcome of a stochastic simulation of this coupled set of equations for $k = 0, \pm 1$ and for two different spin quantum numbers, $S = 10$ and $S = 100$. In these examples it is assumed that the spin is initially prepared in the maximally excited state with $S^z|S\rangle = S|S\rangle$, which we represent by initial distributions as given in Sec. 2.5.1. For the considered values of $S$ we can also solve the full master equation exactly and use these results to benchmark our approximate approach. We find that for about $n_{\text{traj}} = 1000$ trajectories the TWOQS reproduces very accurately the superradiant decay of a large ensemble, with higher accuracy for larger values of $S$. For this example we find almost no visible differences between the three different distribution functions. However, a closer inspection shows that in the case of the Wigner function ($k = 0$), the square root in Eq. (23) can become negative for some trajectories. This becomes a crucial problem for very small values of $S$ and restricts stimulations to short integration times, since at longer times these unphysical trajectories can dominate the dynamics. For larger spins, this error is suppressed by $1/S$ and becomes a negligible effect for $S \gtrsim 100$, as shown in Fig. 1. In a simulation, possible errors arising from the negative diffusion term can be easily tracked by monitoring the change of the total spin, i.e., $\langle|\alpha|^2 + |\beta^2|\rangle$, over time.

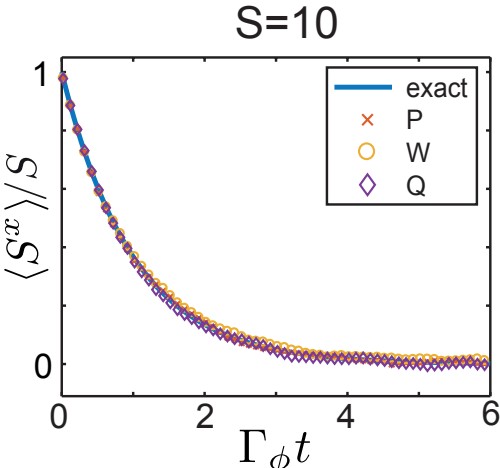
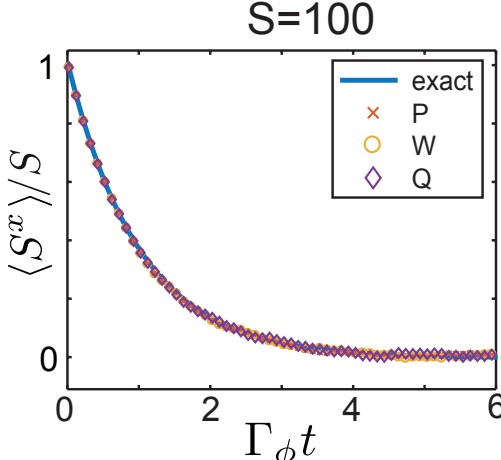

Figure 2: Dephasing of a collective spin as described by Eq. (24). For this plot, it is assumed that the system is initially prepared in a spin coherent state pointing along the $x$-direction, $|S^x = S\rangle$, and the successive evolution of $\langle S^x \rangle(t)$ is shown as a function of time. For this example the other two spin components vanish up to statistical errors. The exact results obtained from the full master equation are compared with stochastic simulations of Eq. (27) and Eq. (28) for $k = 0, \pm 1$. To obtain this data $n_{\text{traj}} = 1000$ trajectories were simulated.

This example illustrates that even for the Wigner function, residual negative diffusion terms are not a practical limitation for simulating dissipative processes in collective spin systems when $S$ is large. Instead, when using the exact positive $P$-representation [31], the same simulation would be limited to times of about $t \lesssim \Gamma^{-1}$, before the appearance of spikes prevents any converging results. Note that the same conclusions also apply to master equations with a gain term, $\mathcal{D}[S^+]$, which can be described by simply exchanging the two bosonic modes, i.e., $\alpha \leftrightarrow \beta$ in Eqs. (22) and (23).

## 3.2 Dephasing

We now proceed with the derivation of the stochastic equations of motion for a collective spin which is subject to dephasing. In the absence of any other interactions, dephasing can be described by the master equation

$$\dot{\rho} = \Gamma_\phi \mathcal{D}[S^z]\rho. \tag{24}$$

The bosonized form of this equation is obtained by substituting $S^z \to (a^\dagger a - b^\dagger b)/2$ and under the TWA the resulting FPE reads

$$\frac{\partial}{\partial t} F_k(\vec{\alpha}, t) = \frac{\Gamma_\phi}{4} \left\{ \frac{\partial}{\partial \alpha}\alpha + \frac{\partial}{\partial \beta}\beta - \frac{\partial^2}{\partial \alpha^2}\alpha^2 - \frac{\partial^2}{\partial \beta^2}\beta^2 + \frac{\partial^2}{\partial \alpha \partial \alpha^*}|\alpha|^2 \right.$$
$$\left. - \frac{\partial^2}{\partial \alpha \partial \beta}\alpha\beta + \frac{\partial^2}{\partial \alpha \partial \beta^*}\alpha\beta^* - \frac{\partial^2}{\partial \beta \partial \beta^*}|\beta|^2 + c.c. \right\} F_k(\vec{\alpha}, t). \tag{25}$$

Although also in this case there are second-order derivatives with negative prefactors, a straight-forward diagonalization of the diffusion matrix shows that $D(\alpha, \beta)$ is already positive semi-definite for all $\alpha$ and $\beta$. In this case the PDA is obsolete and we can factorize the

diffusion matrix as $D(\alpha, \beta) = B(\alpha, \beta)B(\alpha, \beta)^\dagger$, where

$$B(\alpha, \beta) = \sqrt{\frac{\Gamma_\phi}{2}} \frac{i}{4} \begin{pmatrix} \alpha & -\alpha & \alpha & -\alpha \\ -\alpha^* & \alpha^* & -\alpha^* & \alpha^* \\ -\beta & \beta & -\beta & \beta \\ \beta^* & -\beta^* & \beta^* & -\beta^* \end{pmatrix}. \tag{26}$$

Note that this factorization is not unique, but with the current choice we obtain a very simple and symmetric form for the stochastic equations,

$$d\alpha = -\frac{\Gamma_\phi}{4}\alpha dt + i\sqrt{\frac{\Gamma_\phi}{2}}\alpha dW, \tag{27}$$

$$d\beta = -\frac{\Gamma_\phi}{4}\beta dt - i\sqrt{\frac{\Gamma_\phi}{2}}\beta dW, \tag{28}$$

where $dW$ is a single real-valued Wiener processes. These equations are independent of $k$ and there is no preferred phase space distribution to simulate dephasing. In the example plotted in Fig. 2, which shows the dephasing of a spin that is initially polarized along the $x$ direction, the stochastic averages for all distributions agree within the statistical errors with the exact dynamics, keeping in mind that for $k = 1$ and $k = 0$ the initial distributions are only approximate.

## 3.3 Dynamics and Steady States of Driven Spin Systems

We now consider slightly more complicated models in which there is an interplay between coherent driving and incoherent decay. The simplest model in this class is that of a collective spin driven by a transverse field of strength $\Omega$ and including a collective decay with rate $\Gamma$. The corresponding master equations reads

$$\dot{\rho} = -i[H_D, \rho] + \frac{\Gamma}{2S}\mathcal{D}[S^-]\rho, \tag{29}$$

with a Hamiltonian $H_D = \Omega S_x$.

### 3.3.1 Transient dynamics

In Fig. 3 we show again a comparison between the TWOQS and the exact numerical simulations of this master equation for all three distribution functions and for the spin quantum numbers $S = 10$ and $S = 100$. For $S = 10$, we find clearly visible deviations from the exact oscillations, which can in part be traced back to the approximation we made in the initial condition (see Sec. 2.5.1). For this reason, sampling of the $Q$-function is most accurate in this situation. However, these deviations become negligible when we consider higher spins and already for $S = 100$ all distribution functions reproduce very precisely the exact spin dynamics over many oscillation periods.

### 3.3.2 Steady states

A specific interest in the model given in Eq. (29) arises from the fact that it exhibits a non-equilibrium phase transition at a driving strength of $\Omega = \Gamma$ [17, 35, 36]. At this point the

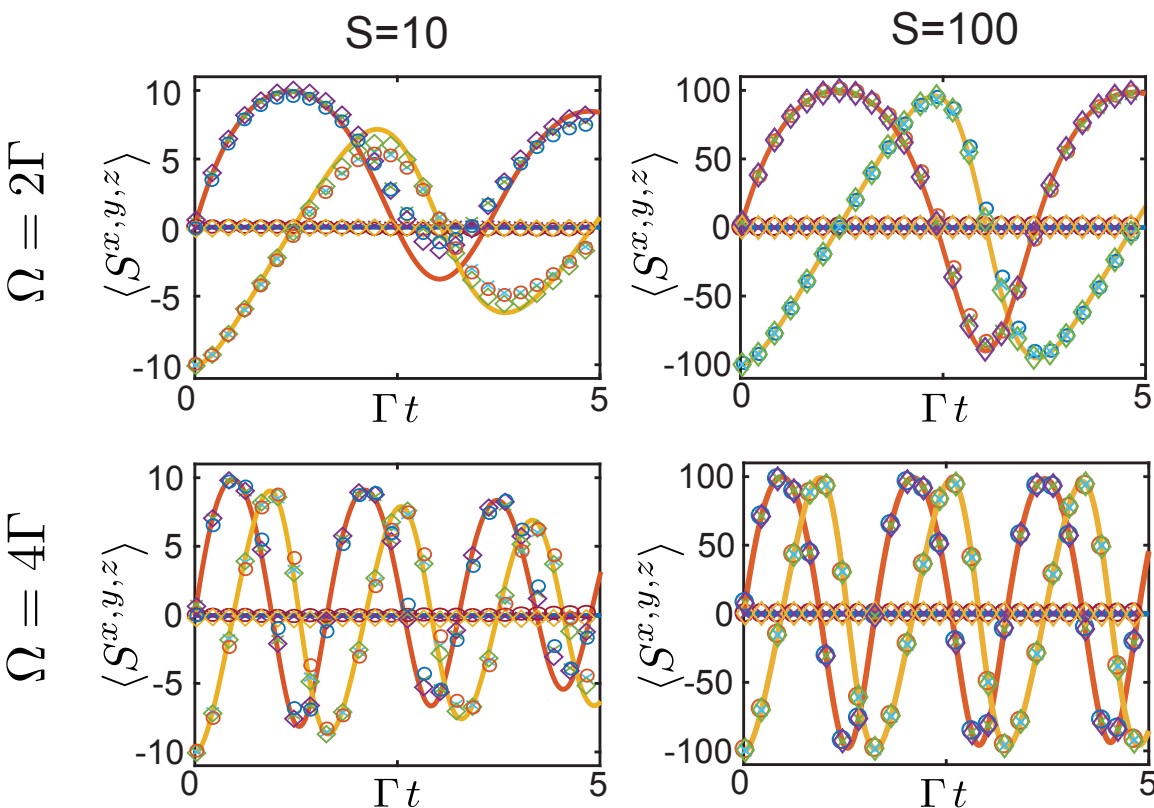

Figure 3: Time evolution of a driven collective spin in the presence of dissipation, as described by Eq. (29). The solid lines represent the exact dynamics of the spin expectation values $\langle S^z \rangle$ (yellow line), $\langle S^y \rangle$ (red line) and $\langle S^x \rangle$ (blue line), while the crosses, diamonds and circles are obtained from the stochastic sampling of the $P$-, the $Q$- and the Wigner-distribution, respectively. For this simulation, the system is initialized in the fully polarized state $|S^z = -S\rangle$.

steady state of this system changes from a spin coherent state on the lower half of the Bloch sphere to a highly mixed state with $\langle S_z \rangle = 0$.

From the analysis of coherent bosonic or spin systems it is known that the TWA often leads to inaccurate results for long simulation times [22]. The same problem is encountered when the TWOQS is used to simulate, for example, the oscillations shown in Fig. 3 for much longer times. However, the timescale beyond which significant errors occur increases with $S$ and for many practical applications the system reaches a steady state before problems arise. This is demonstrated in Fig. 4, where we use our stochastic approach to simulate the master equation for a driven spin with $S = 1000$ up to a time $t = 50\Gamma^{-1}$. Note that for Eq. (29) there still exists an analytic solution for the steady state [17, 35, 36], which allows us to compare these simulations with the exact results for the mean values and the fluctuations of the spin components.

In the polarized phase, $\Omega < \Gamma$, we find that both the mean values as well as the fluctuations of all spin components agree almost perfectly with the exact results. For the considered example of $S = 1000$ there are still some visible differences for the predicted spin fluctuations at and above the transition point, $\Omega/\Gamma = 1$. However, as shown in the inset of Fig. 4(a)

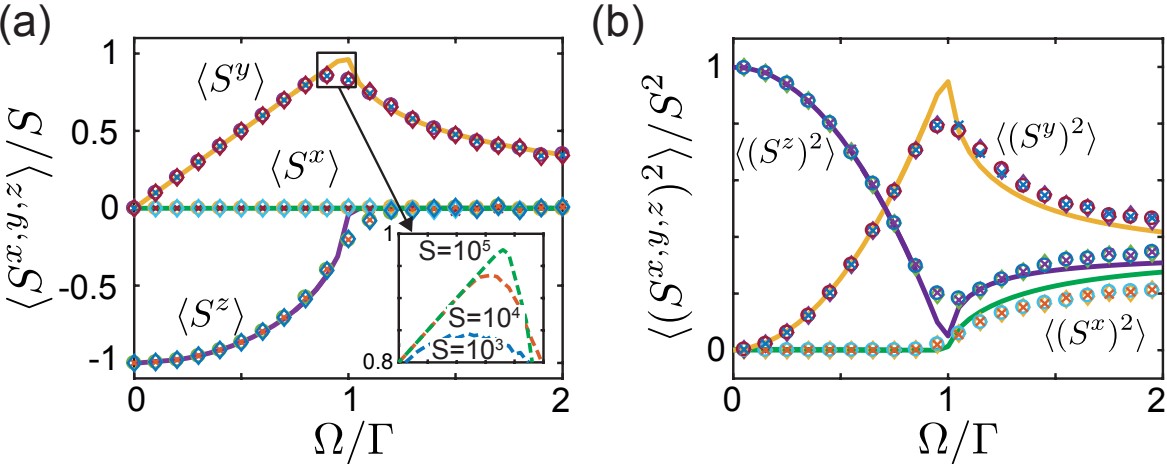

Figure 4: Simulation of the steady state of a driven spin system described by Eq. (29). The two plots show (a) the mean values and (b) the fluctuations of the three components of a spin with $S = 1000$. The solid lines are obtained from the exact solution for the steady state of this system [17, 35, 36], while the crosses, diamonds and circles are obtained from a stochastic sampling of the $P$-, the $Q$- and the Wigner-distribution. The inset in (a) shows the simulations of the Wigner distribution for even larger spin numbers $S$ around the transition point $\Omega/\Gamma = 1$. The steady state was obtained by time averaging after $t = 40\Gamma^{-1}$ for another period of $\Delta t = 10\Gamma^{-1}$ and for $n_{\text{traj}} = 2500$.

the non-analyticity at the phase transition point becomes more pronounced and closer to the exact result by increasing the spin quantum number $S$. We emphasize that in the whole mixed phase, $\Omega/\Gamma \geq 1$, the Liouvillian gap of the considered model, i.e., the smallest decay rate in the problem, scales as $\sim 1/S$. This means that in the mixed phase this system is particularly challenging to simulate and oscillations around the steady state can persist for very long times. Nevertheless, we see that by simply approximating the steady state at a fixed time $t = 40\Gamma^{-1}$ by an average over a time span of $\Delta t = 10\Gamma^{-1}$, all the essential features of the model are already rather accurately reproduced. In particular, for $\Omega \gg \Gamma$, all the fluctuations are around $\langle (S^k)^2 \rangle \sim S^2/3$, indicating that the system is close to a fully mixed state. This and other examples show that by using the TWOQS it is possible to access the steady states of driven-dissipative collective spin models.

## 3.4 Spin Squeezing

Spin squeezing is an important non-classical effect in quantum metrology, which reduces the variance of one spin component below the value of $S/2$ obtained for $\mathcal{N}_{\text{TLS}}$ independent two-level systems. In the presence of collective decay and dephasing, the effect of spin squeezing can be described by the master equation

$$\dot{\rho} = -i\frac{g}{2S}[S_x^2, \rho] + \frac{\Gamma}{2S}\mathcal{D}[S^-]\rho + \Gamma_\phi \mathcal{D}[S^z]\rho, \tag{30}$$

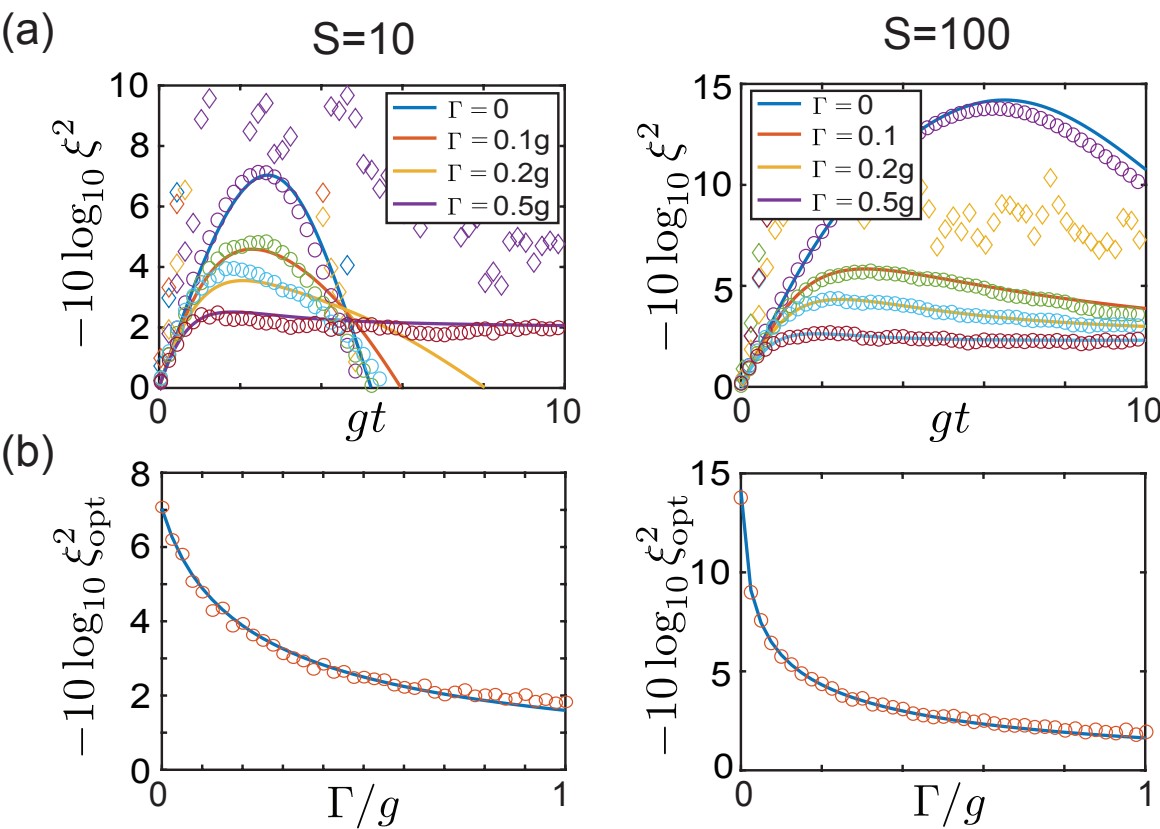

Figure 5: (a) Time evolution of the squeezing parameter $\xi$ for different decay rates $\Gamma/g = 0, 0.125, 0.25, 0.5$ and for $S = 10$ and $S = 100$. (b) Maximum of the squeezing parameter $\xi_{\text{opt}}$ as a function of the decay rate $\Gamma$. In all plots the solid lines represent the exact results, while the diamonds and circles have been obtained from a stochastic sampling of the $Q$- and Wigner distribution. The system is initialized in the state with all spins pointing down, $|S^z = -S\rangle$.

where the Hamiltonian term $\sim S_x^2$ has already been discussed as an example in Sec. 2. Therefore, under the TWA and the PDA we obtain the stochastic equations,

$$d\alpha = -i\frac{g}{4S}\left(\alpha^*\beta^2 + \alpha|\beta|^2\right)dt + d\alpha|_{\text{decay}} + d\alpha|_{\text{deph}}, \tag{31}$$

$$d\beta = -i\frac{g}{4S}\left(\beta^*\alpha^2 + |\alpha|^2\beta\right)dt + d\beta|_{\text{decay}} + d\beta|_{\text{deph}}, \tag{32}$$

where the last two terms in each line account for the decay and dephasing processes described by Eqs. (22)-(23) and Eqs. (27)-(28), respectively.

In Fig. 5, we use the approximate stochastic equations to simulate the spin squeezing parameter $\xi$ as a function of time. For a state pointing in the $z$-direction this parameter is defined as [4]

$$\xi^2 = \min_\phi \frac{2S(\Delta S^\phi)^2}{|\langle S^z\rangle|^2}, \tag{33}$$

where $(\Delta S^\phi)^2 = \langle(S^\phi)^2\rangle - \langle S^\phi\rangle^2$ and $S^\phi = \cos(\phi)S^x + \sin(\phi)S^y$. Note that a squeezing parameter below unity, $\xi < 1$, requires a finite amount of entanglement between the two-level systems [37].

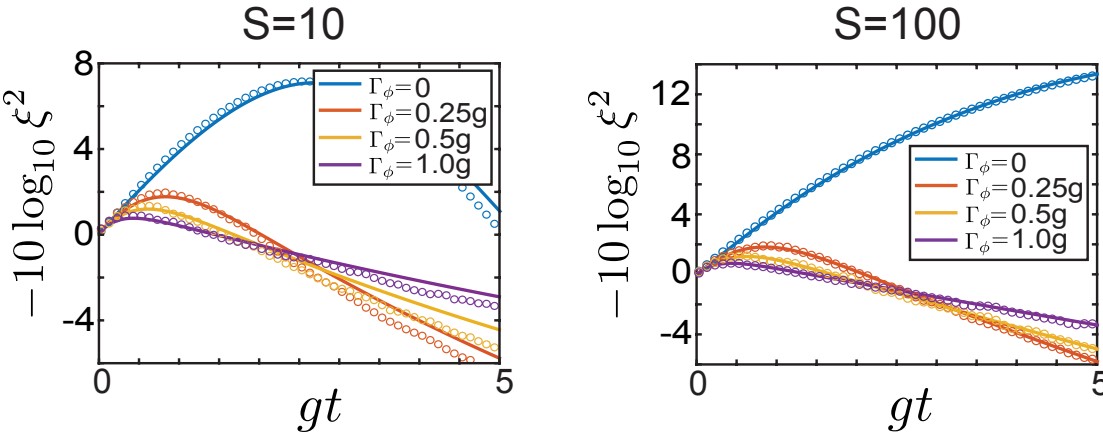

Figure 6: Time evolution of the squeezing parameter $\xi$ for different dephasing rates $\Gamma_\phi/g = 0, 0.25, 0.5, 1.0$ and for $S = 10$ and $S = 100$. The solid lines represent the exact results, while circles have been obtained from a stochastic sampling of the Wigner distribution. The system is initialized in the state with all spins pointing down, $|S^z = -S\rangle$.

Compared to all the previous examples, we now see a clear difference between the results obtained for different distributions. For $k = 1$ the value of the squeezing parameter is $\xi \geq 1$ for all times, since squeezed states can only be represented by a non-positive $P$-distribution. Therefore, these results have not been included in Fig. 5. For the $Q$-distribution we obtain a finite amount of squeezing, but the predicted values for $\xi$ do not match at all the exact results. This discrepancy can be traced back to the fact that in the PDA we neglect essential contributions to the coherent dynamics, which appear whenever there are spin-spin interactions. Therefore, in such cases neither the $P$- nor the $Q$-distribution give reliable predictions.

For simulations based on the Wigner function, we find very accurate results for $\xi$ at short times, but considerable deviations from the exact behavior for longer simulations when $\Gamma$ is small. This is consistent with the general observation that the TWA is not well suited to simulate coherent dynamics over longer times. However, these discrepancies are significantly reduced for larger dissipation rates and for larger spin quantum numbers. Importantly, Fig. 5 shows that already for $S = 100$ the dissipative evolution into an entangled quantum state with $\xi^2 \approx 0.05 - 0.5$ can be accurately simulated with our method. As further demonstrated in the lower two panels of Fig. 5, this level of accuracy is sufficient to predict optimal squeezing parameters in open quantum systems, as relevant for metrological applications. Very similar conclusions can be obtained from the investigation of squeezing in the presence of dephasing, as summarized in Fig. 6. In general we find that dephasing processes are more accurately captured by our method than decay.

## 3.5 Spin Chains

In all the examples so far we have considered the dynamics of a single spin, where for $S \approx 100$ the full master equation can still be solved exactly. This is no longer possible for systems involving $N \gtrsim 2$ collective spins, while the TWOQS scales only linearly with $N$. This feature becomes highly relevant, for example, for the study of non-equilibrium magnetic phases in driven-dissipative spin chains. In this context, one typically considers generic Heisenberg

models of the form [38, 39]

$$H = \sum_{i=1}^{N} \left( \tilde{J}_x S_i^x S_{i+1}^x + \tilde{J}_y S_i^y S_{i+1}^y + \tilde{J}_z S_i^z S_{i+1}^z \right), \tag{34}$$

where in addition each spin is subject to decay. Thus, the master equation for this system reads

$$\dot{\rho} = -i[H, \rho] + \sum_{i=1}^{N} \tilde{\Gamma} \mathcal{D}[S_i^-]\rho, \tag{35}$$

where $\tilde{J}_k = J_k/(2S)$ and $\tilde{\Gamma} = \Gamma/(2S)$ are the rescaled coupling strengths and the rescaled dissipation rate for general spin-$S$ systems.

For $S = 1/2$, Eq. (35) can still be simulated for large 1D chains using numerical techniques based on matrix-product operators [39]. However, in this case one does not observe any sharp phase transitions for finite $\Gamma$, while the reliability and applicability of related techniques for 2D systems are still under investigation [40–42]. Both in 1D and 2D, such tensor network methods have very unfavorable scaling for larger $S$. The current method allows us to address the limit $S \gg 1$, where already in 1D distinct non-equilibrium phases and sharp transitions between them are expected. In a previous work [21] we have already applied this approach to study $\mathcal{PT}$-symmetry breaking transitions in spin chains with both gain and loss, which can be mapped back onto a loss-only model with $\tilde{J}_x = -\tilde{J}_y$ and $\tilde{J}_z = 0$. Here we outline the implementation of this method for the general Heisenberg model in Eq. (34). Since we are dealing with in interacting spin system we must use the Wigner function, i.e., $k = 0$. After carrying out the general procedure described in Sec. 2 we obtain the stochastic equations

$$d\alpha_n = -\frac{i}{4} \left[ (\tilde{J}_x + \tilde{J}_y)(\alpha_{n+1}\beta_{n+1}^* + \alpha_{n-1}\beta_{n-1}^*)\beta_n + (\tilde{J}_x - \tilde{J}_y)(\alpha_{n+1}^*\beta_{n+1} + \alpha_{n-1}^*\beta_{n-1})\beta_n \right.$$
$$\left. + \tilde{J}_z(|\alpha_{n+1}|^2 - |\beta_{n+1}|^2 + |\alpha_{n-1}|^2 - |\beta_{n-1}|^2)\alpha_n \right] dt + d\alpha|_{\text{decay}}, \tag{36}$$

and

$$d\beta_n = -\frac{i}{4} \left[ (\tilde{J}_x + \tilde{J}_y)(\alpha_{n+1}^*\beta_{n+1} + \alpha_{n-1}^*\beta_{n-1})\alpha_n + (\tilde{J}_x - \tilde{J}_y)(\alpha_{n+1}\beta_{n+1}^* + \alpha_{n-1}\beta_{n-1}^*)\alpha_n \right.$$
$$\left. + \tilde{J}_z(|\beta_{n+1}|^2 - |\alpha_{n+1}|^2 + |\beta_{n-1}|^2 - |\alpha_{n-1}|^2)\alpha_n \right] dt + d\beta|_{\text{decay}}. \tag{37}$$

Depending on the relations between all the coupling parameters and the dissipation rate, the model in Eq. (34) exhibits many different stationary phases, which have been analyzed in Ref. [38] using mean-field theory. As a proof-of-concept demonstration of the TWOQS we consider here the case $J_z = 0$. Then for $J_x J_y > -\Gamma^2$ the steady state of the system is the fully polarized state along the $z$-direction and we can use a Holstein-Primakoff approximation to study the fluctuations around this state, similar to the analysis in [21, 38]. Beyond the transition point, e.g. for $J_x > 0$ and $J_y < -\Gamma^2/J_x$, we expect a strongly mixed phase, but in this regime mean-field theory and linearization techniques are no longer applicable. In Fig. 7 we show the results of a stochastic simulation of a spin chain with $N = 100$ sites and $S = 5000$. This simulation confirms that in the limit of large $S$ there is a non-equilibrium phase transition between a polarized and a highly mixed phase, even in 1D. At the transition point the mean value of $\langle S^z \rangle$ and the fluctuations of all spin components exhibit a sharp jump and spin-spin correlations along the chain diverge. In the polarized phase we can still use the

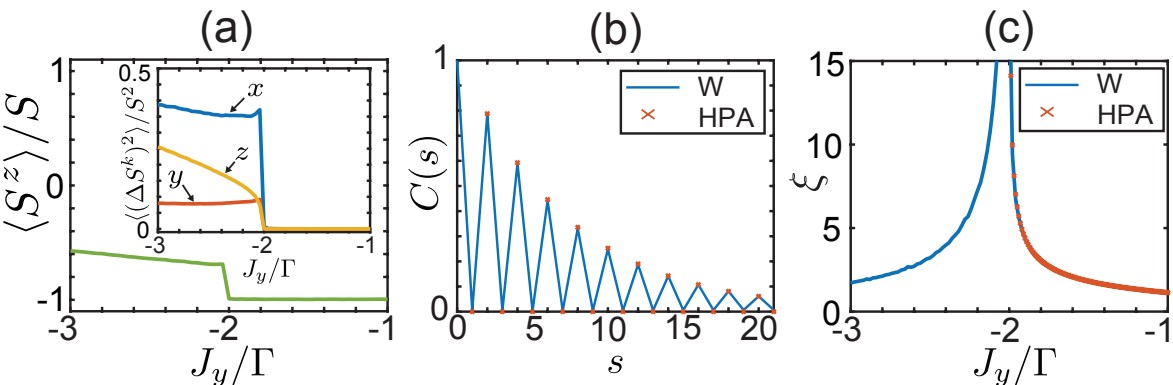

Figure 7: Stochastic simulation of the steady state of a dissipative Heisenberg chain as described by Eq. (35). (a) Magnetization $\langle S^z \rangle$ and variances $\langle (\Delta S^{x,y,z})^2 \rangle$ as a function of $J_y/\Gamma$ for a fixed value of $J_x = \Gamma/2$. (b) Plot of the spin-spin correlations $C(s) = \langle S_n^+ S_{n+s}^- \rangle / \langle S_n^+ S_n^- \rangle$ for a value of $J_y/\Gamma = -1.96$ near the phase transition point. (c) Plot of the correlation length $\xi$ extracted from a fit of $C(s) = e^{-|s|/\xi}$ (for $s$ even) as a function of $J_y$. In all plots the solid line represent the results obtained using the TWOQS and the crosses show the analytic predictions obtain from the Holstein-Primakoff approximation in the polarized phase. For the stochastic simulations we have assumed a chain of $N = 100$ sites with periodic boundary conditions and $S = 5000$.

Holstein-Primakoff approximation to benchmark the simulations also in this extended chain and we find almost perfect agreement. Importantly, the TWOQS also allows us to explore the non-polarized phase, where the strong fluctuations cannot be captured by a Holstein-Primakoff or mean-field approximation. While a detailed analysis of this phase is outside the scope of this work, we find many similarities with the pseudo $\mathcal{PT}$-symmetric phase described in Ref. [21], where further discussions about its physical properties can be found.

## 4   Conclusion

In summary, we have introduced a new numerical method for simulating the dissipative dynamics of collective spin systems. This method works best in the limit of large spin quantum numbers, where exact simulations are no longer possible. At the same time the TWOQS goes beyond mean-field theory by taking the most relevant quantum fluctuations associated with dephasing and decay processes into account. A crucial step in the derivation of the stochastic differential equations is the PDA, which enforces the positivity of the diffusion terms. Although seemingly a very crude approximation, it does not affect the accuracy of actual simulations for large $S$ and allows us to access the long-time dynamics and steady states of open spins systems. This was not possible using previous approaches based on the otherwise more accurate positive $P$-distribution.

   We have illustrated and benchmarked the application of this method for various spin models with dephasing and decay. Since the accuracy of the method improves with increasing $S$ and only scales linearly with the number of spins, these simulations can be directly applied for atomic ensembles with sizes encountered in real experiments or be extended to simulate dissipative spin models in two or even three dimensional lattices. Finally, this technique can

be readily combined with existing TWA simulations for bosonic systems and therefore be applied as well for simulating Dicke-type models, where collective spins are coupled to single or multiple bosonic modes.

**Funding information**   This work was supported through an ESQ fellowship (P.K.) and a DOC Fellowship (J.H.) from the Austrian Academy of Sciences (ÖAW) and by the Austrian Science Fund (FWF) through the DK CoQuS (Grant No. W 1210) and Grant No. P32299 (PHONED).

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
