# Peer review of "Phase-Space Methods for Simulating the Dissipative Many-Body Dynamics of Collective Spin Systems"

_SciPost Physics_

## Round 1 · Referee Report · Anonymous (Referee 1) · 2020-12-23

Strengths

  1. The manuscript describes very clearly an approximate scheme to simulate coupled large spin systems with decay and dephasing described by a Lindblad master equation.
  2. Careful comparison and benchmarking of the method with exact results.
  3. Clearly written and easy accessible manuscript.
  4. The method can be applied in regimes, where conventional approaches fail to be feasible.
  5. Provides access to the full time evolution of the spin system.

Weaknesses

  1. The approximation applied is not well controlled, and it is not a priori possible to judge whether the
  2. The comparison with the exact solution has shown that there are strong deviations at large times if the dissipation is weak. Nevertheless, the authors apply the method afterwards to study a non-equilibrium phase transition at long times for decreasing dissipation. It is not clear, whether the phase transition appears as a breakdown of the method or as a true phase transition.

Report

The manuscript presents a method to simulate the dynamics of large spin systems in the presence of dissipation or decoherence. Such systems are currently of high interests in different area of physics. The method uses several approximations and maps the dynamics to a stochastic equations. Unfortunately, the approximations are not well controlled, however, the comparison with exact results show that in the regime where quantum fluctuations are suppressed by strong dissipation, it gives rise to reliable results. The manuscript is clearly written and present solid results. Therefore, I recommend the manuscript for publication as it is.

Requested changes

Misprint: Below eq. (37), the condition for the appearance of the polarized state and the mixed state is the same. I assume in the first equation it should be > instead of < (i.e., large dissipation).

---

## Round 1 · Referee Report · Anonymous (Referee 2) · 2021-1-19

Strengths

  1. Numerical approximations for open quantum many-body systems are an important field, as these systems are particularly challenging.
  2. The results are novel, especially concerning the application of phase space methods to spin systems.

Weaknesses

  1. The previous literature is presented in an insufficient way.

Report

The manuscript by Huber et al. investigates the use of phase space methods for driven-dissipative spin systems. The key idea is to use a Schwinger boson representation for the spins, which is then tackled using standard phase space methods such as the truncated Wigner approximations. The first parts of the manuscript benchmark the method against exact solutions for collective spin models, while the last part addresses a particularly challenging driven-dissipative Heisenberg model with nearest-neighbor interactions. In principle, this is a potentially interesting work for SciPost Physics, but the authors need to address a couple of points:

  1. The survey on the existing literature for phase space methods for driven-dissipative systems is basically non-existent. Works like [Phys. Rev. B 72, 125335; Phys. Rev. X 5, 041028; Phys. Rev. A 97, 013853] should be referenced somewhere.

  2. The notion of the conserved quantitites of Eq. (1) is not entirely clear. For S = 1/2, the conservation of $\vec{S}_i^2$ appears to be trivial, but that does not help toward reducing the complexity of the problem.

  3. There is a typo in the 3rd line of p.8, where the Q distribution should refer to k = -1.

  4. It is not entirely clear what the authors mean by the P distribution being incapable of representing squeezed states. As the P distribution is an integral transform of the density matrix, the P distribution of squeezed states clearly exists, even if only in a distributional sense. It may be possible that the authors cannot represent such distributions within their numerical work, but then they should say so.

  5. The claim of the method being able to accurately predict the steady state of collective spin models is not justified. Most importantly, the benchmarking against the exact solution seems to fail at the phase transition, i.e., where the dynamics is most interesting. In particularly, the non-analytic behavior of the order parameter does not seem to get captured correctly. Clearly, this point deserves more attention and discussion.

  6. It is not clear what the authors what to say when referring to mean-field theory and how their work constitute a beyond mean-field result. This is particularly striking when comparing with Ref. [33], because the mean-field theory employed there can certainly be applied to highly mixed states (whether it is accurate is another question).

  7. Contrary to what the authors say, there are works on the dissipative Heisenberg model in 2D using tensor network methods [Nature Commun. 8, 1291; arXiv:2012.03095].

Requested changes

see report

---

## Editorial Decision

resubmitted